# Biostimulant and Elicitor Responses to Cricket Frass (*Acheta domesticus*) in Tomato (*Solanum lycopersicum* L.) under Protected Conditions

**DOI:** 10.3390/plants12061327

**Published:** 2023-03-15

**Authors:** Ema Alejandra Ferruzca-Campos, Amanda Kim Rico-Chavez, Ramón Gerardo Guevara-González, Miguel Urrestarazu, Tatiana Pagan Loeiro Cunha-Chiamolera, Rosalía Reynoso-Camacho, Rosario Guzmán-Cruz

**Affiliations:** 1Centro de Investigaciones Aplicadas en Biosistemas (CIAB), Facultad de Ingeniería, Campus Amazcala, Universidad Autónoma de Querétaro, Carr. Chichimequillas-Amazcala Km 1 S/N, Amazcala, Querétaro 76265, Mexico; 2Centro de Investigación en Agrosistemas Intensivos Mediterraneos y Biotecnología Agroalimentaria (CIAIMBITAL), Universidad de Almería, 04120 Almería, Spain; 3Facultad de Química, Universidad Autónoma Querétaro, Cerro de las Campanas S/N, Querétaro 76010, Mexico; 4Cuerpo Académico de Bioingeniería Básica y Aplicada, Facultad de Ingeniería, Campus Amazcala, Universidad Autónoma de Querétaro, Carr. Chichimequillas-Amazcala Km 1 S/N, Amazcala, Querétaro 76265, Mexico

**Keywords:** hormesis, plant growth, biostimulation, elicitation, insect frass

## Abstract

Agriculture in the current century is seeking sustainable tools in order to generate plant production systems with minimal negative environmental impact. In recent years it has been shown that the use of insect frass is an option to be used for this purpose. The present work studied the effect of low doses (0.1, 0.5, and 1.0% *w*/*w*) of cricket frass (*Acheta domesticus*) in the substrate during the cultivation of tomatos under greenhouse conditions. Plant performance and antioxidant enzymatic activities were measured in the study as explicative variables related to plant stress responses in order to determine possible biostimulant or elicitor effects of cricket frass treatments during tomato cultivation under greenhouse conditions. The main findings of this study indicated that tomato plants responded in a dose dependent manner to cricket frass treatments, recalling the hormesis phenomenon. On the one hand, a 0.1% (*w*/*w*) cricket frass treatment showed typical biostimulant features, while on the other hand, 0.5 and 1.0% treatments displayed elicitor effects in tomato plants under evaluated conditions in the present study. These results support the possibility that low doses of cricket frass might be used in tomato cultivation (and perhaps in other crops) for biostimulant/elicitor input into sustainable production systems.

## 1. Introduction

Current agriculture demands plant production systems that mainly use inputs that are not toxic to the environment or to living beings, and that increase crop stress tolerance and their nutritional quality [1]. Any input supporting the abovementioned should be welcomed and included in current agricultural production systems. In the last decade in Europe and for the last five years in the United States of America, a series of natural products derived from cellular components of plants, animals, or microorganisms have been successfully used, known as biostimulants or elicitors. These products, depending on the dose, significantly improve the physiology of the plants for growth and development (biostimulant) and strengthen their immune systems (elicitor) in such a way that the plants display an eustressic behavior. Such biostimulants are viable without having to apply them in the quantities usually used of toxic synthetic agrochemicals, thus producing results with minimal levels of toxicity or even without detected toxicity to the environment or living beings [1]. Biostimulant and elicitor agents that have been used in agriculture are cell extracts (microalgae, plants, etc.), microorganisms that promote plant growth, chemical substances of natural origin such as jasmonic acid, salicylic acid, hydrogen peroxide, fragmented DNA, chitin, and chitosan, among others [1,2,3]. These substances have shown the mentioned effects when applied via foliar or irrigation during cultivation in commercially important species such as chili and tomato [3,4].

The use of biostimulants and elicitors can help to obtain high-quality seedlings [1,2,3,4]. Effects of the application of biostimulants were evaluated—with and without inoculation of *Trichoderma harzianum*—on growth in passion fruit seedlings, where the number of leaves (NH), plant height (AP), chlorophyll index (CI), root length (LR), dry weight of the aerial part (PSPA) and the root part (PSPR) were considered. This study concluded that the application of biostimulants positively affected the biometric NH and PSPR variables in comparison to control [5]. Other studies investigated the effects of the seaweed concentrate ‘Kelpak’ on the growth and mineral nutrition of lettuce plants grown under conditions of varying nutrient supply. Kelpak significantly increased the yield, concentration, and amounts of Ca, K, and Mg in the leaves of lettuce, achieving an adequate supply of nutrients [6].

Thus, bioproducts containing live microorganisms or natural compounds derived from organisms, such as bacteria, fungi, and algae, improve plant growth and restore soil fertility [7]. Another alternative is to evaluate formulations derived from chitin (and other carriers of microorganisms beneficial to plant growth) that are also used as biostimulants in agriculture [8]. Chitin is obtained by recycling exoskeleton waste from crustacean shrimp, crabs, and crickets [9]. The most widely used derivative in agriculture is chitosan, the N-deacetylation of chitin obtained under alkaline conditions [10,11]. The beneficial effects of chitosan have been previously reported to promote plant growth and increase tolerance to biotic and abiotic stress [12]. In recent years, the use of insects as food has led to an increase in the number of private companies dedicated to the production of insect-derived products [13]. An essential part of the mass-rearing process is the production of frass (insect excreta) by insects, which supposes an important end product within the system that might be considered an organic fertilizer and food for livestock farms; although under-exploited, such frass is has possibilities for use as agricultural input [14,15].

The present research aimed to evaluate the biostimulant and elicitor potential of frass derived from the intensive cultivation of crickets (*Acheta domesticus*) in a substrate for the production of tomato (*Solanum lycopersicum*) under protected conditions. Some morphological characteristics such as plant height, basal stem diameter, and leaf number, as well as some antioxidant enzyme producers such as superoxide dismutase (SOD), catalase (CAT), and phenylalanine ammonium lyase (PAL), all related to plant stress responses, were evaluated as plant immunity markers. Our results displayed that depending on the dose, the cricket frass amendment in the substrate displayed a hormetic curve behavior in the evaluated variables. Treatment 0.1% (*w*/*w*) showed biostimulant features, while 0.5 and 1.0% treatments displayed elicitor effects in tomato plants under evaluated conditions. The results suggested the possibility of proposing a new product for sustainable production agriculture based on cricket frass and give value to a residue from the production of crickets for human consumption.

## 2. Results

### 2.1. Effect of Cricket Frass on Plant Morphological Variables in Tomato

The results obtained from the morphological variables for each treatment and control is shown in Figure 1. As observed, it demonstrated a linear growth of the morphological variables during the crop development evaluated. On the one hand, it can be seen that the treatment in which the substrate is composed of 99.9% sand and 0.1% frass displayed a biostimulant effect. The former asseveration was based on the results of the tomato plants in morphological variables related to an eustressic plant performance, with significantly higher height, basal diameter, and number of leaves compared to the control. On the other hand, the treatments where the percentage of frass was 0.5% and 1%, showed a likely elicitor effect since the plants reached a significantly lower height, basal diameter, and number of leaves, thus trending to a distressic behavior compared to the control (Figure 1); however, no apparent visual symptoms of toxicity were observed.

The Analysis of Variance for the morphological variables of the plants reaffirmed observations regarding differences in treatment among plants in terms of their heights, the basal diameter of their stems, and the number of their leaves. Therefore, the Tukey Test was performed to compare all possible combinations of pairs of means, and it was found that there were statistically significant differences among the treatments that these means represent (Table 1).

The effects on morphological characteristics of tomato plants shown by cricket frass applications evaluated suggest that depending on the dose, the effect might be that of a biostimulant (0.1% frass), displaying a higher behavior in the three morphological variables evaluated as shown in the regression study displayed in Figure 1 and Table 1. Moreover, frass doses of 0.5 and 1% showed decreased behavior in the same analysis compared to the control, likely suggesting that instead of plant biomass production, the plant is allocating energy for immunity to cope with stress conditions (elicitation), although not significantly affecting plant performance.

### 2.2. Effect of Cricket Frass on Plant Antioxidant Enzymatic Immunity Markers

In order to characterize in more detail the response of tomato plants to different cricket frass doses, the Analysis of Variance determined that commonly evaluated antioxidant enzymes be considered plant immunity markers. All data are presented as the mean ± SD (Figure 2). The evaluated enzymes related to stress response were phenylalanine ammonia lyase (PAL), catalase (CAT) and superoxide dismutase (SOD), the former being a key enzyme in the phenylpropanoids (PPDs) synthesis, while CAT and SOD participate in the scavenging of reactive oxygen species (ROS) during plant stress responses [16]. The results shown in Figure 2 display, on the one hand, that in comparison to control, treatment with 0.1% frass displayed a significant decrease for PAL activity, as well as a significant increase in CAT activity; SOD activity was not different in comparison to control. On the other hand, the treatment with 1% frass showed the highest decrease and increase for PAL and CAT activities, respectively, with no difference in SOD in comparison to the control (Figure 2). These results together suggest that treatments with 0.5% and 1% frass caused a decrease in phenylpropanoids production. The resources saved in this task are probably being used for plant growth in the case of the biostimulant treatment (0.1% frass), and preferentially to produce another type of secondary metabolite different from PPDs (i.e., terpenes, alkaloids) related to defense instead of to plant growth in the case of elicitor treatment (1% frass).

Moreover, in the present research, it was found that the use of cricket frass in low doses (0.1–1%) within the substrate displayed a hormetic effect such as biostimulant or elicitor behavior in the tomato plant’s morphological and biochemical variables related to the plant stress response being evaluated (Figure 3).

## 3. Discussion

Although the abovementioned results showed that cricket frass improved plant performance and, probably, the elicitation features in the tomato plants evaluated, this behavior depended on the week of the measurement (Figure 1 and Table 1). Changes in morphological characteristics displayed a trend to behave like control plants. Similar results were obtained in applying elicitation to chili peppers, a phenomenon in which the plant’s developmental stage might be explained regarding the response to external stress factors applied to the plants during cultivation [16].

Taken together, the effects on morphological characteristics and antioxidant enzyme activities of tomato plants shown by the evaluated cricket frass applications suggested that depending on the dose, the plants might lean toward eustressic or distressic behavior. This indicates the possibility that the hormetic phenomenon is participating in the experimental model evaluated. Biostimulant frass (0.1%) displayed improved behavior in the morphological variables evaluated. On the other hand, frass doses of 0.5 and 1% showed worsened behavior in the same analysis compared to the control, likely suggesting that instead of plant biomass production, the plant is allocating energy for immunity to cope with stress conditions (elicitation), although this does not significantly affect plant performance (Figure 4).

Both biostimulation and elicitation are positive plant responses that, based on the hormesis phenomenon, can be defined as part of the eustress zone in the biphasic hormetic curve [1,17]. The case of elicitation is interesting to highlight because if it is excessive, the plant will be in distress, thus showing symptoms of toxicity as suggested above [1,18]. It is likely that the level of nutrients (or possibly another component not analyzed in the frass, i.e., levels of chitin residues) that cricket frass contains (Table 2) provides a type of “alert cue” in the plants that also participate in inducing the plant stress response observed at morphological levels.

As the results suggest, the treatments with 0.5% and 1% of cricket frass in the substrate caused a decrease in the production of phenylpropanoids. Probably the resources saved in this task are being used for plant growth, as occurs in the case of the 0.1% frass treatment (biostimulant), and preferably to produce other types of secondary metabolites different from PPDs related to defense instead of to growth in the treatment of 1% frass (elicitor). The latter asseveration suggests that depending on the dose, the cricket frass is causing the trade-off commonly reported in the application of stress factors to plants, provoking hormesis [16]. The CAT activity results with both frass treatments also support the abovementioned, because the treatment with the highest CAT activity (1% frass) suggests high H_2_O_2_ (ROS) detoxification activity that is provoking elicitation in such a way that the hormetic behavior is related to the distress zone; while for treatment with 0.1% frass, although significantly higher in comparison to control, is likely generating an H_2_O_2_ level in the range of the eustress zone in a hormetic curve related to biostimulation [19]. The fact that SOD activity was not different from control in both frass treatments suggests that the H_2_O_2_ (ROS) production as stress response caused by frass in the tomato plants evaluated in this work, is perhaps mainly caused by other enzymes, such as plasma membrane NADPH oxidases, peroxisonal oxidases, type III peroxidases, and other apoplastic oxidases [20].

Recent studies by other authors using insect frass (Hexafrass^TM^) showed that depending on the plant variable evaluated, a hormetic effect was observed; even a dose of 6 g per pot of chicory and ribwort showed high toxicity on shoot growth and yield [21]. Although our study lasted a short time (60 days) in relation to the complete cultivation cycle of tomatos, the results showed that based on the hormetic responses observed, it would be possible to evaluate in future work the possibility that this frass residue could be evaluated in complete cultivation cycles to determine its effects on fruit yield and quality, as well as on plant protection against pest and diseases—and even as a fertilizer source, based on its composition shown in Table 2. These future studies should consider the fact that insect frass might also have a positive effect on plants because it is a vector of beneficial microorganisms (not evaluated in the present work). Ref [22] conducted a study using mealworm frass, which found an increase in tolerance to biotic and abiotic stress of seedlings against drought, floods, and salinity, due to the sterilization of excrement, identifying numerous bacterial and fungal isolates capable of fixing atmospheric nitrogen, solubilizing phosphates and potassium, and producing siderophores, auxins, and 1-aminocyclopropane-1-carboxylic acid (ACC) deaminase. Ref. [23] found that the excreta of different herbivorous insects were capable of activating the defensive responses of the plant mediated by salicylic acid and jasmonic acid. Ref. [24] found that the microorganisms present in insect droppings can activate the defensive responses of the *S. frugiperda* corn plant, since the bacterium *Pantoea ananatis* was isolated and identified in insect droppings, demonstrating its ability to increase the expression in the plant of the gene that codes for the inhibitor of the proteinase of maize induced by herbivores (mpi), which causes a decrease in attacks of the insect.

## 4. Materials and Methods

### 4.1. Plant Growth Conditions and Fertigation

The experiment was carried out in controlled growth under a greenhouse of 30 m^2^ located in Amazcala, Querétaro (Mexico). On 15 April 2022, seedlings were transplanted into individual 500-mL containers filled with coconut fiber substrate, whose physicochemical characteristics were described by [25]. The experiment ended on 15 June 2022. During the experimentation within the greenhouse, the temperature was 25 to 18 °C/18 to 16 °C (day/night, respectively) with a relative humidity of 80 to 85%. The nutrient solution was based on a standard nutrient solution of [26]. The pH of the nutrient solutions was always maintained at 5.8 with the addition of nitric acid; the amount of nitric acid required always increased nitrate to a negligible extent. New fertigation was applied when the water in the crop unit reached 10% of the easily available water [27,28,29].

### 4.2. Frass Nutrient Characteristics

The levels of macro and micronutrients and the levels of secondary elements were determined in a commercial laboratory (Fertilab-Celaya, México) using standard methods for each chemical element. The results and the method that the commercial laboratory uses to measure each chemical element are shown in Table 2.

### 4.3. Treatments

For the study, treatments corresponding to mixtures of inert substrate (sand): frass (*w*/*w*) in the proportions 99:1, 99.5:0.5, and 99.9:0.1 were carried out; as the control, 100% sand was evaluated. Twenty-four seedlings of the Saladette type tomato were transplanted in triplicate on these mixtures with 6–8 true leaves per each treatment (n = 24). Plants were located within bags with 3 kg of each treatment, and the study used a planting density of 2 plants m^2^ distributed in a random block design in the greenhouse (8 plants per block). The plants were grown in the greenhouse. Watering with a nutrient solution irrigated the substrate every 2 d with 300 mL of nutrient solution. The nutrient solution was prepared using highly soluble fertilizer salts to obtain the next ion concentration: 12 meq L^−1^ of NO_3_^−^; 1 meq L^−1^ of H_2_PO_4_^−^; 7 meq·L^−1^ of SO_4_^−2^; 7 meq L^−1^ of K^+^; 9 meq L^−1^ of Ca^+2^; and 4 meq L^−1^ of Mg^+2^. The pH of the nutrient solution was kept at 6 ± 0.2, and the electrical conductivity was kept between 1.6 and 1.8 dS mL^−1^. The production system lasted two months in the greenhouse.

### 4.4. Determination of Plant Morphological Characteristics

Morphometric measurements of height, basal stem diameter, and leaf number in the plants were taken weekly. The plant height was measured using a ruler from the base to the tip of the shoot, the basal stem diameter was measured with a digital vernier, and the numbers of leaves were counted according to [30].

### 4.5. Antioxidant Enzyme Activities

#### 4.5.1. Superoxide Dismutase (SOD) Enzyme Activity

Samples consisting of 9 leaves from 3 different plants from each block were taken to perform biochemical antioxidant enzyme activity analyses for each treatment. SOD activity was determined by a modified method according to [31]. The plant material was thoroughly ground with liquid nitrogen; then 0.3 g of plant powder was separately homogenized in 2 mL of 50 mM phosphate buffer (pH 7.8) at 4 °C. The homogenate was vortexed for 2 min and then centrifuged for 15 min at 12,000 rpm min^−1^ at 4 °C. The supernatant was the enzyme extract used to determine the enzyme activity. Test tubes were added, each containing: 1.5 mL of phosphate buffer (pH 7.8), 0.3 mL of 0.1 mM EDTA-Na_2_, 0.3 mL of 0.13 M methionine, 0.3 mL of 0.02 riboflavin, 0.3 mL of 0.75 mM nitroblue tetrazolium (NBT), and 0.25 mL distilled water. The reaction was initiated by adding 0.05 mL enzyme extract to each tube and incubating for 30 min at 25 °C under white light exposure. Finally, each absorbance was recorded at 560 nm against a blank. One unit of SOD was expressed as the amount of protein that caused a 50% reduction of NBT in the reaction, and the enzyme activity was expressed as unit mg^−1^ protein.

#### 4.5.2. Catalase (CAT) Enzyme Activity

Catalase activity was determined according to the method reported by [32] with some modifications. The basis was the decrease in absorbance at 240 nm caused by the decomposition of H_2_O_2_ by the catalase enzyme. The enzyme extract was obtained by homogenizing 0.3 g of frozen plant material in 2 mL of a 100 mM Tris-HCl buffer (pH 8) containing 30 mM 2-mercaptoethanol and 20% (*v*/*v*) glycerol at 4 °C. The homogenate was vortexed for 2 min and centrifuged for 15 min at 12,000 rpm at 4 °C. The supernatant was the enzyme extract used for the activity test. For measuring CAT activity, 0.1 mL of the enzyme extract was added to test tubes containing 2 mL of 50 mM phosphate buffer and 0.2 mL of 100 mM H_2_O_2_. The change in absorbance of the reaction mixture was determined by measuring the absorbance every minute for 6 min. The CAT activity was calculated as the H_2_O_2_ extinction coefficient of 0.0392 mM cm^−1^ and expressed as mmol H_2_O_2_ mg^−1^ protein.

#### 4.5.3. Phenylalanine Ammonium Lyase (PAL) Activity

PAL activity was determined by the method reported by [33] with some modifications. First, 0.3 g of each frozen sample was homogenized in 2 mL of 0.1 M sodium borate buffer containing 0.1% 2-mercaptoethanol (pH 8.8) at 4 °C. The extract was vortexed for 2 min and then centrifuged at 12,000 rpm for 15 min at 4 °C. The supernatant was the enzyme extract for the activity test. PAL activity was analyzed as the ratio of conversion of L-phenylalanine to trans-cinnamic acid at 290 nm in a UV–Vis spectrophotometer. Samples containing 0.2 mL of the enzyme extract were added to 2.3 mL of sodium borate buffer (pH 8.8) containing 10 mM L-phenylalanine. The reaction mixture was incubated at 40 °C for 60 min. After the incubation time, the reaction was forced to stop by adding 0.5 mL of 1 N HCl and left for 10 min. The absorbance was measured at 290 nm, and the values were compared to a 7-point trans-cinnamic acid calibration curve.

### 4.6. Statistical Analysis

For the statistical analysis, the Analysis of Variance (ANOVA) was performed. With the ANOVA, each of the morphological and biochemical variables were measured in various groups of tomato plants, each of which receives a different dose of cricket frass: 0% (control), 0.1%, 0.5%, and 1%. The Tukey’s HDS post hoc test was performed for each experiment to determine statistical differences using the software STATGRAPHICS XV. A *p* ˂ 0.05 was considered significant.

## 5. Conclusions

Taken together, our results suggest that cricket frass under the evaluated conditions in the present work, when less than 1% is used in the substrate, displayed either a biostimulant (0.1%) or elicitor (0.5 and 1.0%) effect on tomato crops. Both the biostimulant or elicitor effects were reflected in the morphology and antioxidant enzyme activities evaluated, showing improved features in relation to the control. Future research studying insect frass, in addition to increasing insight into using it as a biostimulant/elicitor in other crops as sustainable agricultural input, should also include research into the possibility of the frass being used as fertilizer, based on the essential chemical elements composition it displays.

## Figures and Tables

**Figure 1 plants-12-01327-f001:**
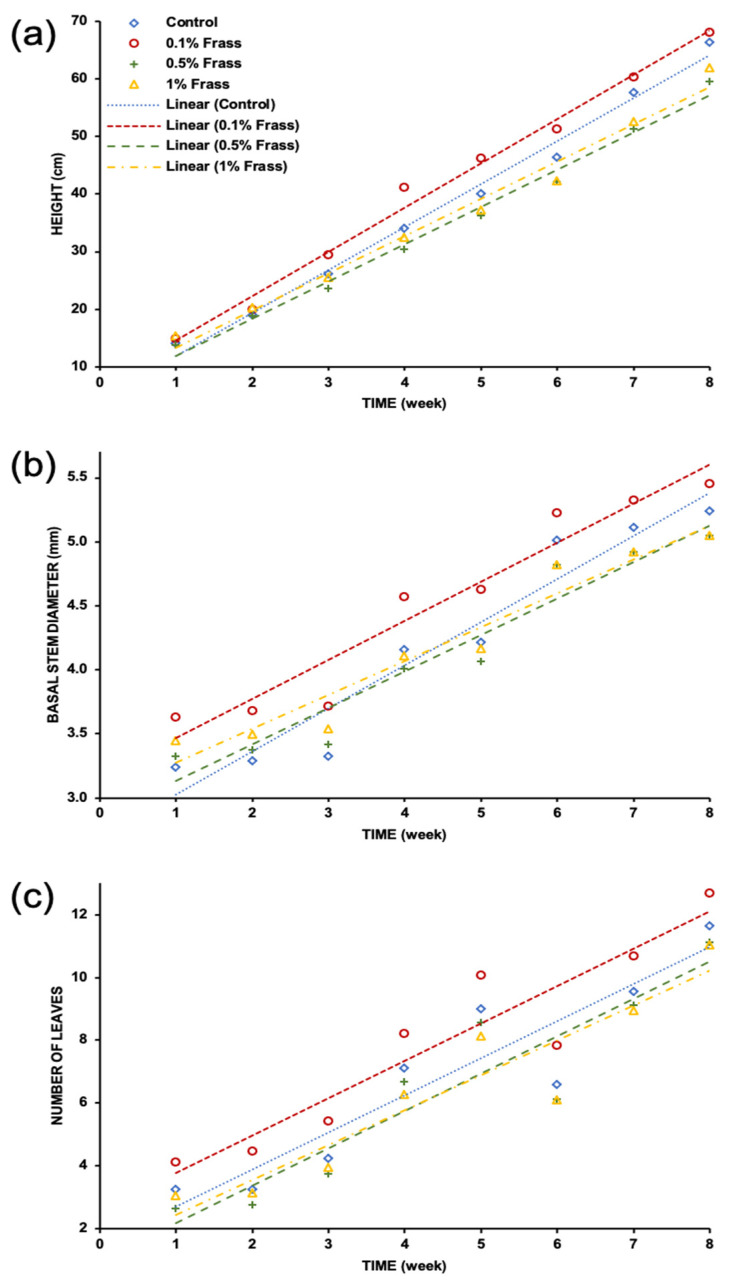
Dispersion diagram of morphological variables (**a**) height, (**b**) basal stem diameter, and (**c**) the number of leaves.

**Figure 2 plants-12-01327-f002:**
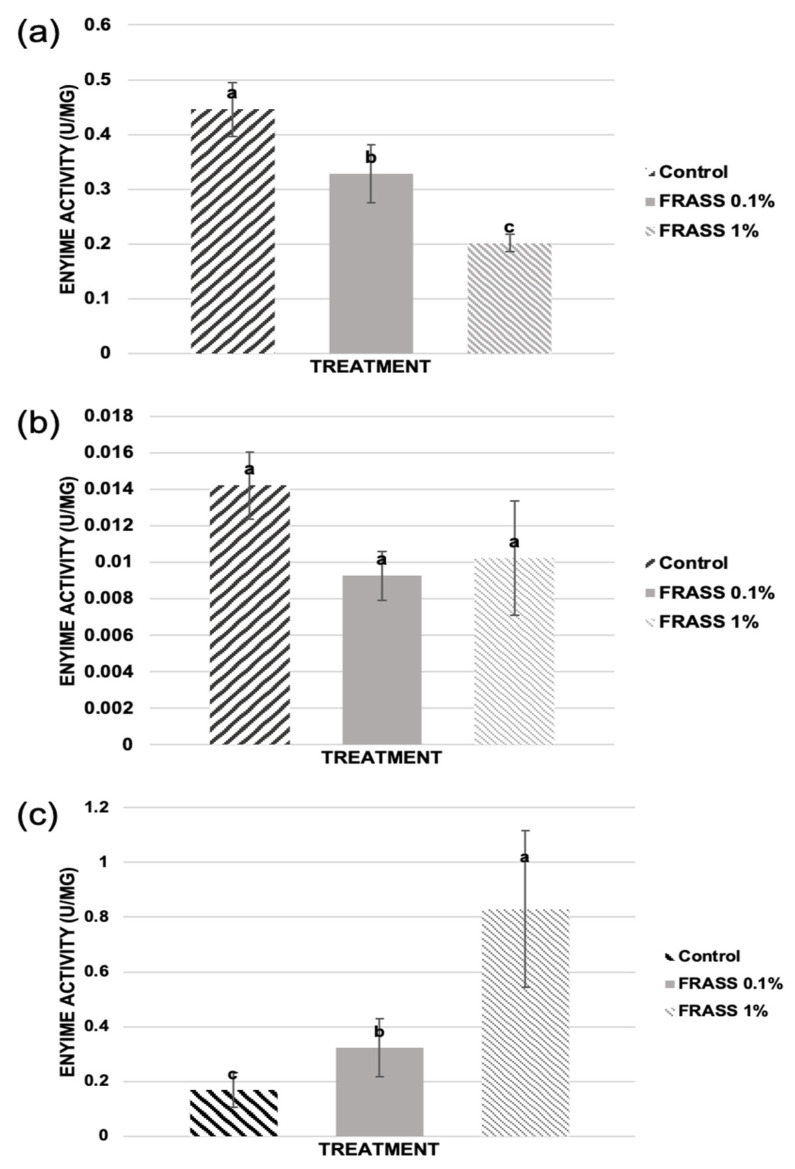
Effect of cricket frass on the (**a**) PAL, (**b**) SOD and (**c**) CAT enzyme activities in tomato plants. Different letters in each histogram for each enzyme indicate a statistically significant difference after two-way ANOVA (*p* < 0.05) and Tukey’s HDS test for multiple comparisons (95%). Error bars represent the standard deviation of the mean.

**Figure 3 plants-12-01327-f003:**
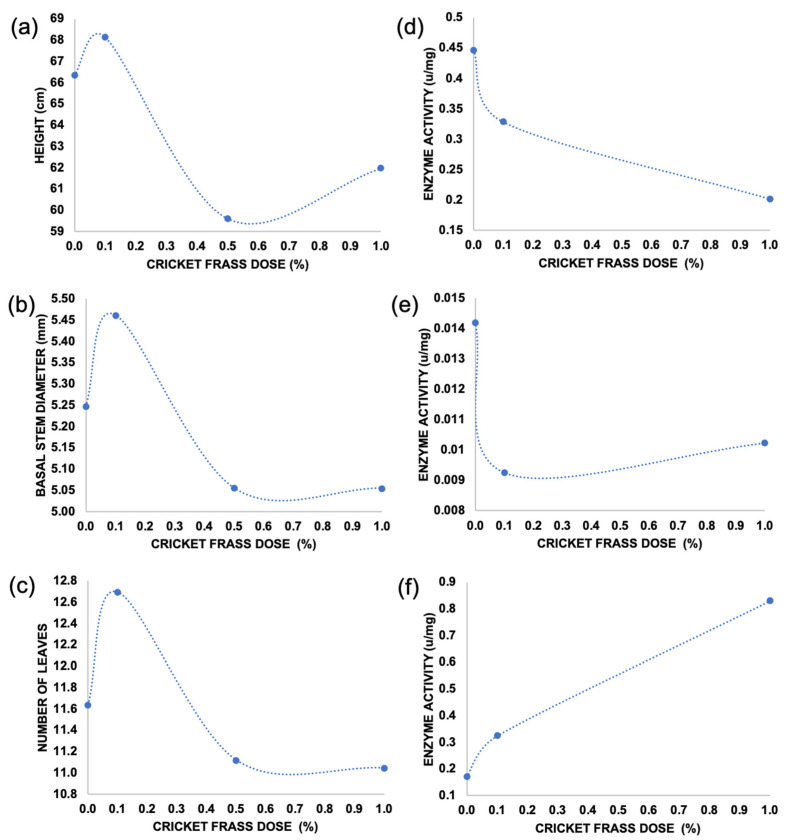
Dispersion diagram of morphological variables (**a**) height, (**b**) basal stem diameter, and (**c**) the number of leaves; and biochemical enzyme activities variables (**d**) PAL, (**e**) SOD and (**f**) CAT.

**Figure 4 plants-12-01327-f004:**
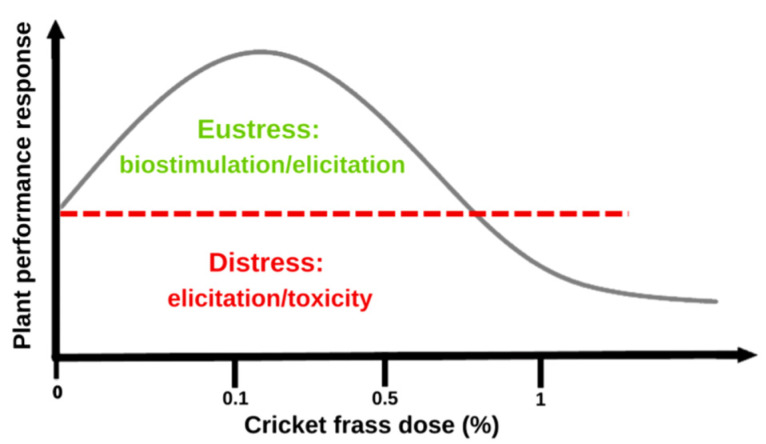
Graphical representation of the hormetic effect observed in tomato treated with frass in the substrate in the present study.

**Table 1 plants-12-01327-t001:** Morphological response of tomato plants to the addition of cricket Frass to the cultivation substrate.

Morphologic Variables	Week	Control	0.1% Frass	0.5% Frass	1% Frass
Height	1	14 ^a^	15 ^a^	14 ^a^	15 ^a^
	2	19 ^a^	20 ^a^	19 ^a^	20 ^a^
	3	26 ^a^	29 ^a^	24 ^a^	26 ^a^
	4	34 ^a,b^	41 ^a^	32 ^b^	32 ^b^
	5	40 ^a,b^	46 ^a^	38 ^a,b^	37 ^b^
	6	44 ^a,b^	51 ^a^	42 ^b^	42 ^b^
	7	58 ^a^	60 ^a^	51 ^a^	53 ^a^
	8	66 ^a^	68 ^a^	60 ^a^	62 ^a^
Stem diameter	1	3.24 ^a^	3.64 ^a^	3.33 ^a^	3.45 ^a^
	2	3.29 ^a^	3.69 ^a^	3.38 ^a^	3.50 ^a^
	3	3.33 ^a^	3.73 ^a^	3.42 ^a^	3.54 ^a^
	4	4.16 ^a,b^	4.58 ^a^	4.02 ^b^	4.11 ^b^
	5	4.22 ^a,b^	4.64 ^a^	4.07 ^b^	4.18 ^ab^
	6	5.02 ^a^	5.23 ^a^	4.82 ^a^	4.82 ^a^
	7	5.12 ^a^	5.33 ^a^	4.92 ^a^	4.92 ^a^
	8	5.25 ^a^	5.46 ^a^	5.05 ^a^	5.05 ^a^
Number of leaves	1	3 ^a^	4 ^a^	3 ^a^	3 ^a^
	2	3 ^a^	4 ^a^	3 ^a^	3 ^a^
	3	4 ^a^	5 ^a^	4 ^a^	4 ^a^
	4	7 ^a,b^	8 ^a^	6 ^b^	6 ^b^
	5	9 ^a,b^	10 ^a^	8 ^b^	8 ^b^
	6	7 ^a^	8 ^a^	6 ^a^	6 ^a^
	7	10 ^a^	11 ^a^	9 ^a^	9 ^a^
	8	12 ^a^	13 ^a^	11 ^a^	11 ^a^

Different letters in each row (week of each treatment) indicate a statistically significant difference after a two-way ANOVA (*p* < 0.05) and a Tukey’s HDS test for multiple comparisons (95%).

**Table 2 plants-12-01327-t002:** Nutrient analysis of the cricket frass used in the study.

Nutrients	Element	Units	Results	Method
Macronutrients	Total nitrogen	%	4.035	MicroKjeldahl
	Total Phosphorus (P_2_O_5_)	%	1.560	Spectrophotometric
	Potassium (K_2_O)	%	1.820	Atomic absorption
Secondary elements	Calcium	%	1.340	Atomic absorption
	Magnesium	%	0.510	Atomic absorption
	Sulfur	%	0.699	Turbidimetric
	Sodium	%	0.620	Atomic absorption
Micronutrients	Iron	ppm	334.040	Atomic absorption
	Cooper	ppm	47.770	Atomic absorption
	Manganese	ppm	154.350	Atomic absorption
	Zinc	ppm	195.680	Atomic absorption

## Data Availability

Not applicable.

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
