# Peer review of "Biostimulant and Elicitor Responses to Cricket Frass (Acheta domesticus) in Tomato (Solanum lycopersicum L.) under Protected Conditions"

_plants, 2023, doi:10.3390/plants12061327_

Round 1

Reviewer 1 Report

This article presented Biostimulant and elicitor responses to cricket frass (Acheta domesticus) in tomato (Solanum lycopersicum L.) under protected conditions. Before recommending this article for publication, the article need substantial major revisions.

Replace the words “practices in century XXI are looking” with “in the current century or 21st

Proper methods are not described in the abstract. The authors should add brief methods in the abstract.

Main findings must be presented in the abstract.

Line 43-46 the sentence is very long and confusing should be revise.

Line 64-65 add recent reference as well. The following study will be helpful. https://doi.org/10.3390/molecules27196281

Line 70 should be cited with relevant study https://doi.org/10.1016/j.bcab.2020.101729

Line 83-84 italicize the plant names.

The introduction section is very long, many sentences or purpose of the sentences is repeating in the intro. These should be revise.

Why only plant physiological and enzyme characterization was performed. There are no biochemical tests in this study. Provide the reason in discussion or introduction section.

Elaborate the methods of section 2.2

“were taken weekly. Morphometric measurements of the plants were taken weekly” check the repetition

Replace “the number of leaves was” was with were” Line 135

Examination of chlorophyll contents would be better in this study. Why these parameters are missing.

Conclusion should be results based presenting future recommendations and research gaps.

Reviewer 2 Report

The manuscript deals with the evaluation of different doses of cricket frass extract as biostimulant or elicitor on tomato plants. Authors have measured morphological traits and enzymatic activities related to stresses.

Te manuscript is well structured. The used methords seem clearly descried. Statistical analyses, even not sufficient replications, are sounds.

results are presented according a mechanistc approach. which intersting.

The manuscript needs, however, several modifications.

1-there is no comparison with already published works.

2-some parts of introduction should be documented (adding recent references).

3- lease follow the authors guidelines of Plants. The M&M section should be placed après discussion and befroe conclusion.

See

https://www.mdpi.com/journal/plants/instructions#preparation

Minor changes

Abstract line 21. Scientific name should be in italic

L52 The use of biostimulants and elicitors can help to obtain high-quality seedlings. Please add a reference.

L54 Trichoderma harzianum shoul be in italic

L70-72. Please add more recent references. See for examples few references.

https://doi.org/10.3390/polym15040866

https://doi.org/10.3390/microorganisms11020467

https://doi.org/10.3390/polym15051099

Please follow the authors guidelines of Plants. The M&M section should be placed après discussion and befroe conclusion.

See

https://www.mdpi.com/journal/plants/instructions#preparation
